

# Effects of salinity change on two superoxide dismutases (SODs) in juvenile marbled eel *Anguilla marmorata*

Li Wang[*], Xiaolu Wang[*] and Shaowu Yin

Jiangsu Key Laboratory for Biodiversity and Biotechnology, College of Life Sciences, Nanjing Normal University, Nanjing, Jiangsu, China
Co-Innovation Center for Marine Bio-Industry Technology of Jiangsu Province, Lianyungang, Jiangsu, China
[*] These authors contributed equally to this work.

## ABSTRACT

Salinity is one of the most important factors that affect the fish growth and survival. Superoxide dismutases (SODs), as the primary antioxidant enzymes, play a first role in the process of preventing oxidative stress caused by excessive superoxide anion ($O_2^-$) in living organisms. In the present study, we investigated the effects of salinity on the gene expressions as well as enzymatic activities of MnSOD and Cu/ZnSOD in gill, intestine, kidney, liver and muscle tissues of the marbled eel *Anguilla marmorata*. We found that the liver might possess stronger redox capacity compared with other tissues. Furthermore, the gene expressions and enzymatic activities of SODs in juvenile marbled eels could be effectively enhanced by low salinity but inhibited when the salinity was higher than the body tolerance. Our findings indicated that MnSOD and Cu/ZnSOD played vital roles in the adaptation of marbled eels to salinity variation, which contributed to the elucidation of physiological adaptation and regulatory mechanism of SODs in eels.

## INTRODUCTION

Salinity is one of the most important factors that affect fish growth and survival, since its variation may cause a series of physiological stress responses in aquatic animals, leading to imbalance of serum hormone levels, energy metabolism and electrolytes (*Choi, An & An, 2008*). Recent studies have shown that stress responses caused by salinity variations are closely associated with enhanced generation of reactive oxygen species (ROS) (*Livingstone, 2001*). However, excessive ROS can lead to oxidative stress and cell malfunction, finally resulting in the apoptosis or necrosis (*Hermes-Lima & Zenteno-Savin, 2002*; *Sun et al., 2014*). Organisms have developed defense mechanisms to shield themselves from such oxidative damage (*Marikovsky et al., 2003*). Superoxide dismutase (SOD) is a key enzyme that can prevent oxidative stress through catalyzing the dismutation reaction of superoxide anion ($O_2^-$) into $O_2$ and $H_2O_2$ in living organisms ($H_2O_2$ is subsequently transformed into $H_2O$ by catalase) (*Vaughan, 1997*). SODs (EC 1.15.1.1) can be classified into four distinct

Corresponding author
Shaowu Yin,
yinshaowu@hotmail.com

groups based on their structures, cellular localizations and metal cofactors at their active sites: copper/zinc SOD (Cu/ZnSOD), manganese SOD (MnSOD), iron SOD (FeSOD) and nickel SOD (NiSOD) (*Bannister, Bannister & Rotilio, 1987*; *Kim et al., 1998*).

Each type of SODs shows distinct genomic- and proteomic-structural characteristics and subcellular distributions. Usually, Cu/ZnSOD and MnSOD are localized in cytoplasm and mitochondrial matrix, respectively. Cu/ZnSOD serves as a bulk scavenger of radicals in the intracellular environment (*Chakravarthy et al., 2012*), and MnSOD (*Tian et al., 2011*) plays a key antioxidant role in mitochondria (*Cho et al., 2009*). Previous studies have demonstrated that SOD expression is modulated by endotoxins (*Cho et al., 2009*; *Sook Chung et al., 2012*), pathogens (*Tian et al., 2011*; *Yu et al., 2011*) and environmental pollution (*Lopes et al., 2001*), suggesting a critical role of SODs in antioxidant system. In recent years, much attention has been paid to the connection between salinity and antioxidant responses of fish (*Ransberry et al., 2015*; *Yin et al., 2011*). The study on marine fish *Pampus argenteus* (*Yin et al., 2011*) showed that certain low salinity can activate SOD, but its activity may be inhibited as the salinity drops below its tolerance range. Moreover, in fish *Pseudosciaena crocea* (*Wang et al., 2015*), SOD activity in the kidney is increased with reduction in salinity within a range from 7‰ to 28‰. However, most investigations in fish encountering salinity changes have focused on the changes in the activities of antioxidant enzymes; nevertheless, less attention has been paid to the transcriptional level.

Marbled eel *Anguilla marmorata* belongs to Osteichthyes, and it is one of the quintessential tropical catadromous fishes. This fish live widely across tropical and subtropical oceans and are associated with fresh water (FW) systems. *A. marmorata* has been on the International Union for Conservation of Nature Red List of Threatened Species due to over fishing and environmental pollution, and it is regarded as species under the second-class national protection in China (*Wang et al., 2014*). The life cycle of *A. marmorata* includes five stages as follows: leptocephalus, glass eel, elver, yellow eel and silver eel, while *A. marmorata* must migrate from sea water (SW) to FW for growth and development from the stage of elver (*Li et al., 2015*; *Lin et al., 2012*). Although previous studies have shown that the antioxidant enzyme activity can be altered by salinity changes (*Yin et al., 2011*; *Wang et al., 2015*), the regulatory mechanism of SODs in salinity adaptation of eels remains poorly understood.

In this study, we identified two SODs, denoted as AmMnSOD and AmCu/ZnSOD. Moreover, we assessed their mRNA expression levels in eels in FW and analyzed the temporal mRNA expression profiles and enzymatic activity *in vivo* after they were transferred to brackish water (BW) and SW. Our results provided comparative perspectives into the two widespread and functional diverse enzymes, and offered important evidence to clarify the physiological adaptation and regulatory mechanism of SODs in eels.

## MATERIALS AND METHODS

Juvenile *A. marmorata* ($18 \pm 0.81$ cm in length, $18 \pm 0.77$ g in weight) from FW were collected from Wenchang, Hainan Province, China by Hainan Wenchang Jinshan Eel Technology Co., Ltd. This company has obtained the People's Republic of China aquatic

wild animal catching permit from Ministry of Agriculture of The People's Republic of China since 2004 (Approval number: National Fishery Resources and Environmental Protection 2004; 13). This study was also approved by the Ethics Committee of Experimental Animals at Nanjing Normal University (Research permit number: NNU20120301). All the eels were transferred to the tanks filled with filtered FW in the laboratory and fed to satiation with a commercial feed for eels every day. After acclimation at 25–26 °C for 1 week, they were used for the challenge experiments.

## Salinity treatment and tissue sampling

The eels were divided into a control group and two experimental groups. The experimental groups consisted of BW (salinity of 10‰) and SW (salinity of 25‰) groups. In the control group, eels were reared in FW (salinity of 0‰). In BW and SW groups ($n = 72$ for each group), the eels were primarily placed in FW, and then the salinity was gradually increased by 3‰ every day until it reached BW or SW. To evaluate the mRNA expression of two AmSODs under normal physiological condition, multiple tissues, including brain, gill, spleen, intestine, liver, kidney, muscle and heart, were collected from six eels in the control group. In order to determine the defense responses of AmSODs in these salinity adapted groups, multiple tissues, including gill, intestine, liver, kidney and muscle, were collected from six eels in the experimental groups at 1 h, 3 h, 6 h, 12 h, 1 and 2 d after the desired salinity was established. During the sampling process, experimental eels were anaesthetized with a solution of 0.05% MS-222 (Sigma, USA). In addition, the collected samples were also used to determine enzymatic activity of SODs. During the experimental period, salinity and pH (6.5–7.5) were monitored daily.

## Total RNA extraction and cDNA synthesis

Total RNA was extracted from above-mentioned tissues using High Purity RNA Fast Extract Reagent (BioTeke, Beijing, China) according to the manufacturer's instructions, and extracted RNA was stored at −80 °C before further analysis. The RNA concentration was determined using NanoDrop 2000 (Thermo, Wilmington, DE, USA), and its integrity was examined on 1.0% agarose gel. The single-strand cDNA was synthesized using HiScript$^{TM}$ QRT SuperMix (Vazyme, Piscataway, NJ, USA) for subsequent quantitative real-time PCR (qRT-PCR).

## Analysis of AmSOD expression

In our previous study, the full-length AmMnSODs and AmCu/ZnSOD have been cloned using the 3′ and 5′ rapid amplification cDNA end (RACE) method, and their NCBI accession numbers are KR350467 and KR350468, respectively (*Wang et al., 2016*). Tissue distribution and temporal expression profiles of AmMnSOD and AmCu/ZnSOD in eels under normal conditions (FW group) and eels with salinity treatment (BW and SW groups) were investigated by qRT-PCR. Table 1 lists all the gene-specific primers for AmMnSOD, AmCu/ZnSOD and Am$\beta$-actin used in this study. The experiments were performed in a 20-μL reaction system consisting of 4 μL of diluted cDNA template, 10 μL of Faststart Universal SYBR Green Master (Roche, Basel, Switzerland), 1 μL of each primer (6 mmol/μL) and 4 μL ddH$_2$O, and each experiment was performed in

**Table 1** List of primers used in this study.

| Primer name | Purpose | Primer sequence (5′-3′) |
| --- | --- | --- |
| CuZnSOD-F | qRT-PCR amplification | CTTCAACCCGCACAACAAGA |
| CuZnSOD-R | qRT-PCR amplification | TGCCGGTTTTCAAGCTTTCA |
| MnSOD-F | qRT-PCR amplification | CACAGCAAACACCACGCC |
| MnSOD-R | qRT-PCR amplification | TGGACATCTTCTCCCTCAGC |
| Amβ-actin-F | qRT-PCR internal reference | GCAGATGTGGATCAGCAAGC |
| Amβ-actin-R | qRT-PCR internal reference | ACATTGCCGTCACCTTCATGC |

triplicate. Briefly, after a denaturation step at 94 °C for 10 min, the amplification was carried out with 40 cycles at a melting temperature of 94 °C for 10 s, an annealing temperature of 55 °C for 30 s, and an extension temperature of 72 °C for 60 s. A melting curve was generated after each reaction to confirm the efficiency of qRT-PCR, and absence of primer dimers or other non-specific products was also verified based on the analysis of the melting curve. The relative expression level of AmSOD transcripts was determined by the $2^{-\Delta\Delta Ct}$ comparative Ct method using $\beta$-actin as an internal control (*Livak & Schmittgen, 2001*). The calculated relative expression level of AmSODs in each tissue was compared with its respective level in spleen in the tissue-specific expression analysis. In the time-course analysis, the fold-change post salinity treatment was determined by comparing with the expression level in FW group.

## Measurement of SOD enzymatic activities

The SOD enzymatic activity in the above-mentioned tissues was determined using the SOD Typing Testing Kit (Jiancheng Bioengineering, Nanjing, China) after the salinity reached the target salinity. The experimental tissues were homogenized in normal saline (0.85% (w/v) of NaCl, denoted as NS) by an electric homogenizer. Coomassie Brilliant Blue was used to determine the protein concentration in the crude extract according to the manufacturer's protocol (Jiancheng Bioengineering, Nanjing, China). The total SOD enzymatic activity and Cu/ZnSOD activity were determined following the manufacturer's instructions. Each sample was measured in triplicate.

## Statistical analysis

All data were expressed as mean $\pm$ SD of triplicates, and the results were subjected to one-way analysis of variance (one-way ANOVA) and two-tailed paired $t$ test with SPSS v17.0 software. A difference was considered to be statistically significant at $P < 0.05$ and extremely significant at $P < 0.01$.

## RESULTS

### Spatial expression and tissue distribution of AmSODs

Figure 1 shows that the mRNA expression levels of both SODs were detectable in all the eight tissues examined by qRT-PCR assay, but their relative baseline expression levels varied. Predominant expression of AmMnSOD was detected in liver, muscle and heart tissues

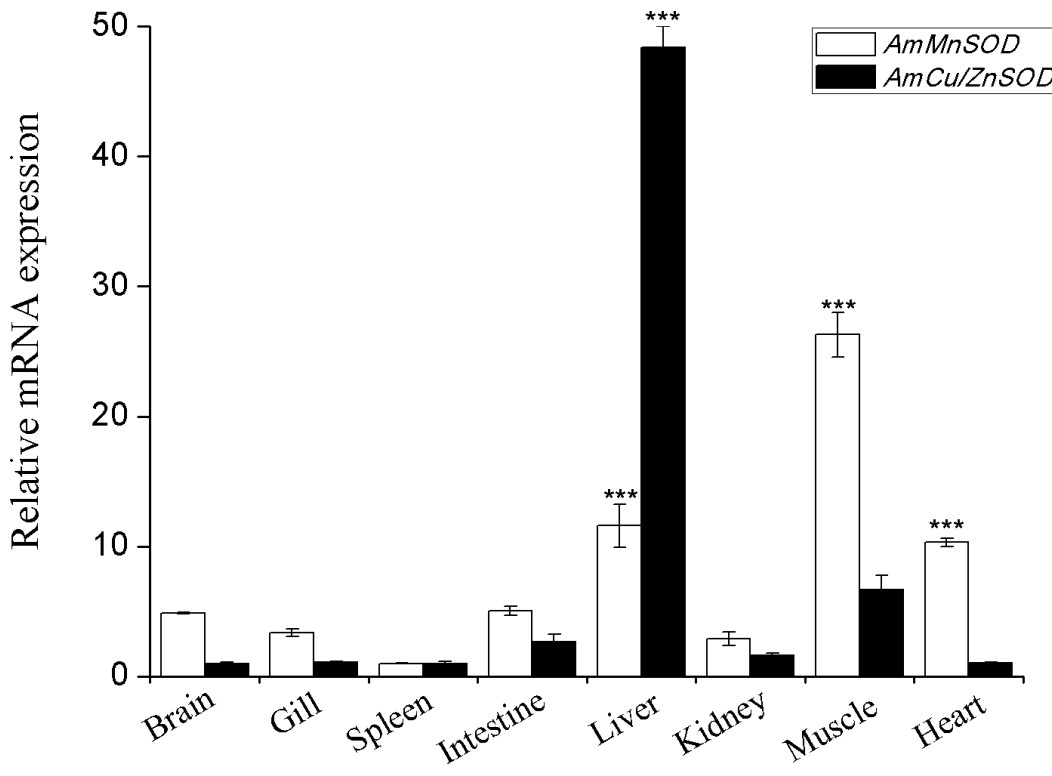

**Figure 1** **Tissue distribution analysis of AmMnSOD and AmCu/ZnSOD at the mRNA level of**
**_A. marmorata._** The relative mRNA expression levels of AmSODs in each tissue were calculated by the
$2^{-\Delta\Delta Ct}$ method using _A. marmorata β-actin_ as an internal reference gene. Vertical bars represent the
S.D. ($n = 3$). Data indicated with asterisk symbol (*) are significantly different from the spleen tissue,
* at $P < 0.05$, ** at $P < 0.01$ and *** at $P < 0.001$.

($P < 0.001$; 11.6-fold, 26.3-fold and 10.3-fold, respectively). In contrast, AmCu/ZnSOD
was highly expressed in liver ($P < 0.001$; 48.4-fold). Moreover, the mRNA abundance of
AmSODs was low in other tissues.

## Temporal transcriptional regulation of the two AmSODs

Although no mortality or pathologies during the experiment, our results clearly revealed
that the salinity variation significantly altered the expressions of AmSODs. In gill, kidney,
liver and muscle tissues (Figs. 2, 3A and 3C–3E), the expression levels of AmMnSOD
and AmCu/ZnSOD showed a trend of rising at first and then reducing with prolonged
time in BW. However, the expression levels of AmMnSOD and AmCu/ZnSOD were
first decreased and then exhibited an upward trend in gill, liver and muscle tissues in
SW. However, the expression level of AmCu/ZnSOD was first increased and decreased
afterwards in kidney in SW. In contrast, the expression level of AmMnSOD was significantly
greater than that of the control group at 12 h and 2 d ($P < 0.001$) merely. In the intestine
(Figs. 2B and 3B), the mRNA levels of AmMnSOD and AmCu/ZnSOD were barely changed
within 6 h in both BW and SW, and then both reached their peak levels at 12 h in SW
($P < 0.001$). However, the expressions of AmMnSOD and AmCu/ZnSOD peaked at 1 d
and 12 h ($P < 0.01$) in BW, respectively. In addition, the expression levels of AmMnSOD in
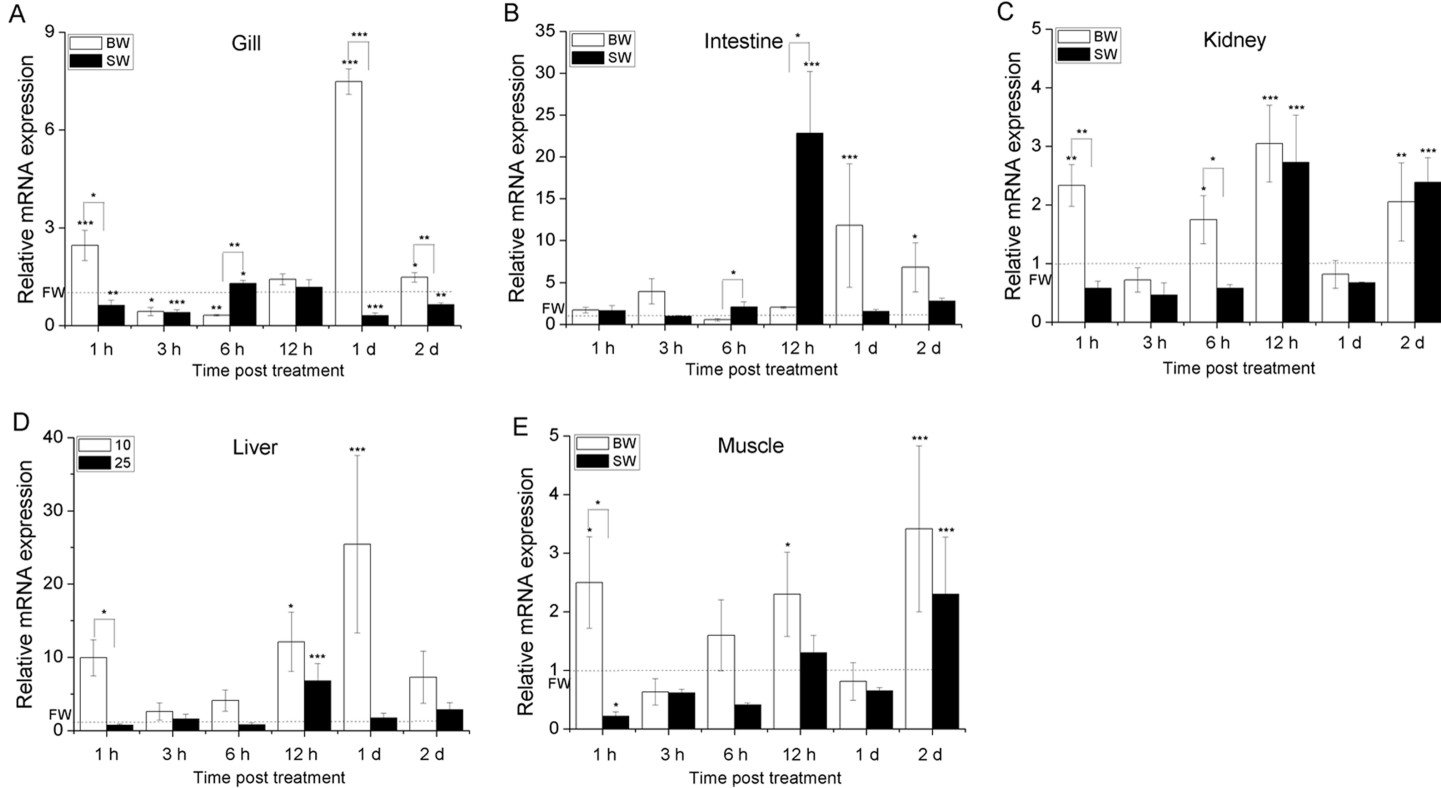

**Figure 2** Temporal mRNA expression analysis of AmMnSOD in gill (A), intestine (B), kidney (C), liver (D) and muscle tissues (E) of juvenile eels due to the change of salinity. The relative mRNA expression level of AmMnSOD in each tissue was determined using *A. marmorata β-actin* as an internal reference gene. Then the expression levels of AmMnSOD were presented as fold-change relative to FW. Vertical bars represent the S.D. ($n = 3$). Data indicated with asterisk symbol (*) are significantly different from the corresponding FW control and data with asterisk symbol (*) on the box indicated significant difference between BW and SW, * at $P < 0.05$, ** at $P < 0.01$ and *** at $P < 0.001$.

intestine and AmCu/ZnSOD in kidney were significant higher in SW compared with BW. In comparison, the expression levels of AmMnSOD in gill and AmCu/ZnSOD in muscle were significant higher at 6 h and 2 d in SW compared with BW, respectively. However, their expression levels were significant lower at 1 d in SW compared with BW. Moreover, the expression levels of AmMnSOD (kidney, liver and muscle) and AmCu/ZnSOD (gill, intestine and liver) were significant higher in BW compared with SW.

## Changes in SOD enzymatic activities

In order to examine the antioxidant status in *A. marmorata* in response to different salinity levels, we determined the AmSOD enzymatic activities in different treatment groups.

Figures 4 and 5 show that the variation trend of total SOD activity was similar to that of Cu/ZnSOD activity in intestine, kidney and muscle tissues. However, a significantly different variation trend was observed between the total SOD activity and Cu/ZnSOD activity at 2 d in gill and liver tissues of BW group (Figs. 4, 5A and 5D). Moreover, the Cu/ZnSOD activity in gill was significantly decreased at first and then increased in BW, while the Cu/ZnSOD activity in intestine, kidney, liver and muscle tissues was first increased and then decreased. Figures 5A, 5C and 5E show that the changes of Cu/ZnSOD activities
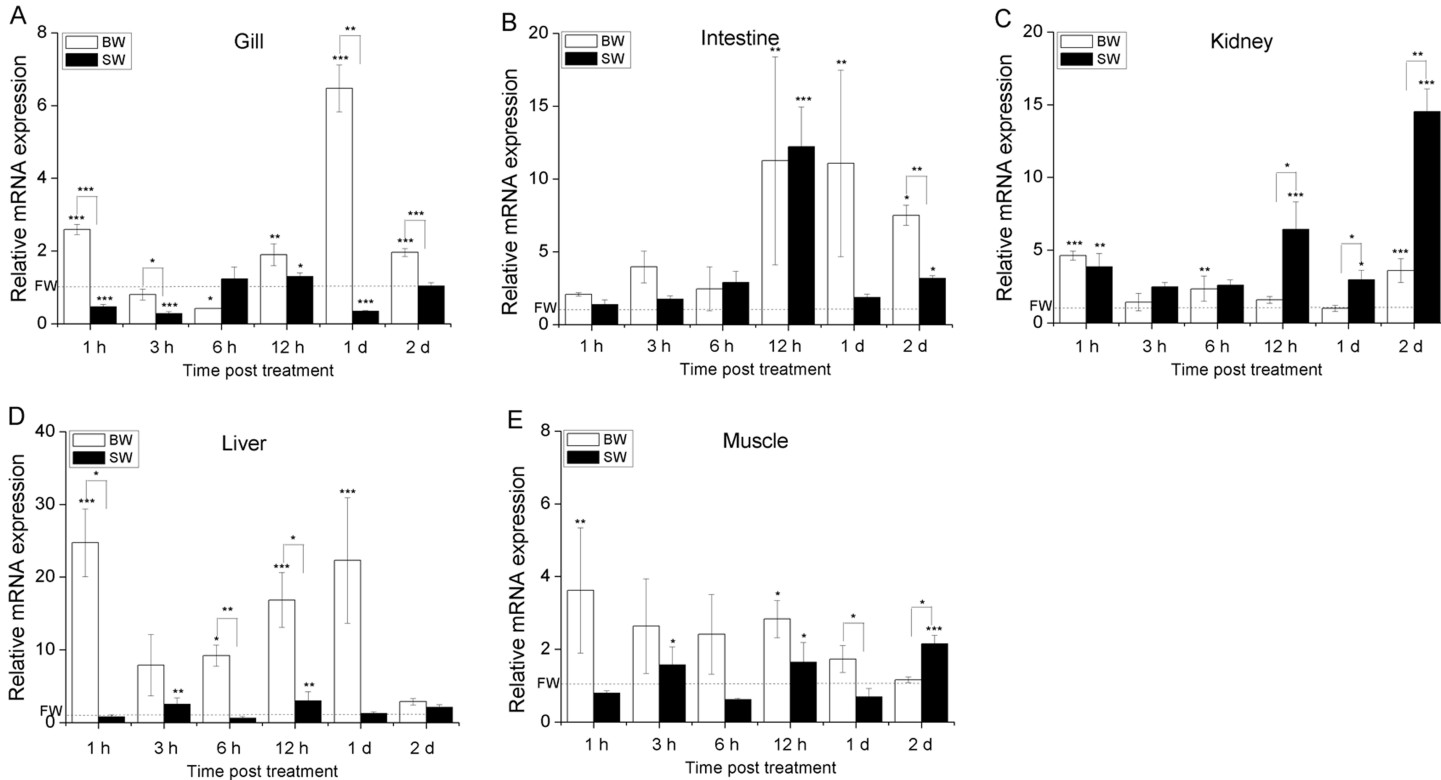

**Figure 3** **Temporal mRNA expression analysis of AmCu/ZnSOD in gill (A), intestine (B), kidney (C), liver (D) and muscle tissues (E) of juvenile eels due to the change of salinity.** The relative mRNA expression level of AmCu/ZnSOD in each tissue was determined using *A. marmorata β-actin* as an internal reference gene. Then the expression levels of AmCu/ZnSOD were presented as fold-change relative to FW. Vertical bars represent the S.D. ($n = 3$). Data indicated with asterisk symbol (*) are significantly different from the corresponding FW control and data with asterisk symbol (*) on the box indicated significant difference between BW and SW, * at $P < 0.05$, ** at $P < 0.01$ and *** at $P < 0.001$.

were similar in gill, kidney and muscle tissues in SW, exhibiting an overall decreasing trend within 2 d. Figure 5B shows that the Cu/ZnSOD activity in intestine was maintained at the basal level ($P < 0.05$). Conversely, the Cu/ZnSOD activity was distinctively altered in liver, showing a significant up-regulation from 1 h to 1 d ($P < 0.05$) in SW (Fig. 5D). In addition, the total SOD and Cu/ZnSOD activities in kidney and muscle tissues exhibited an overall higher level within 2 d in BW compared with SW. In contrast, the total SOD and Cu/ZnSOD activities in other tissues in BW were only higher at several time points compared with SW, such as in gill and intestine tissues. Moreover, the total SOD activity in liver showed a lower level within 2 d in BW compared with SW.

## DISCUSSION

All the experimental fish were denoted as juvenile eel by measuring their body weight and length. Therefore, they should be more easily threatened by salinity pressure compared with adult *A. marmorata*. In addition, our preliminary experiment also revealed the rapid salinity increase of water environment may lead to death of experimental fish, and the similar results have been reported in other fish species, such as *Oreochromis mossambicus* (*Li et al., 2014*). Previous study has indicated that *A. marmorata* must adapt to three types

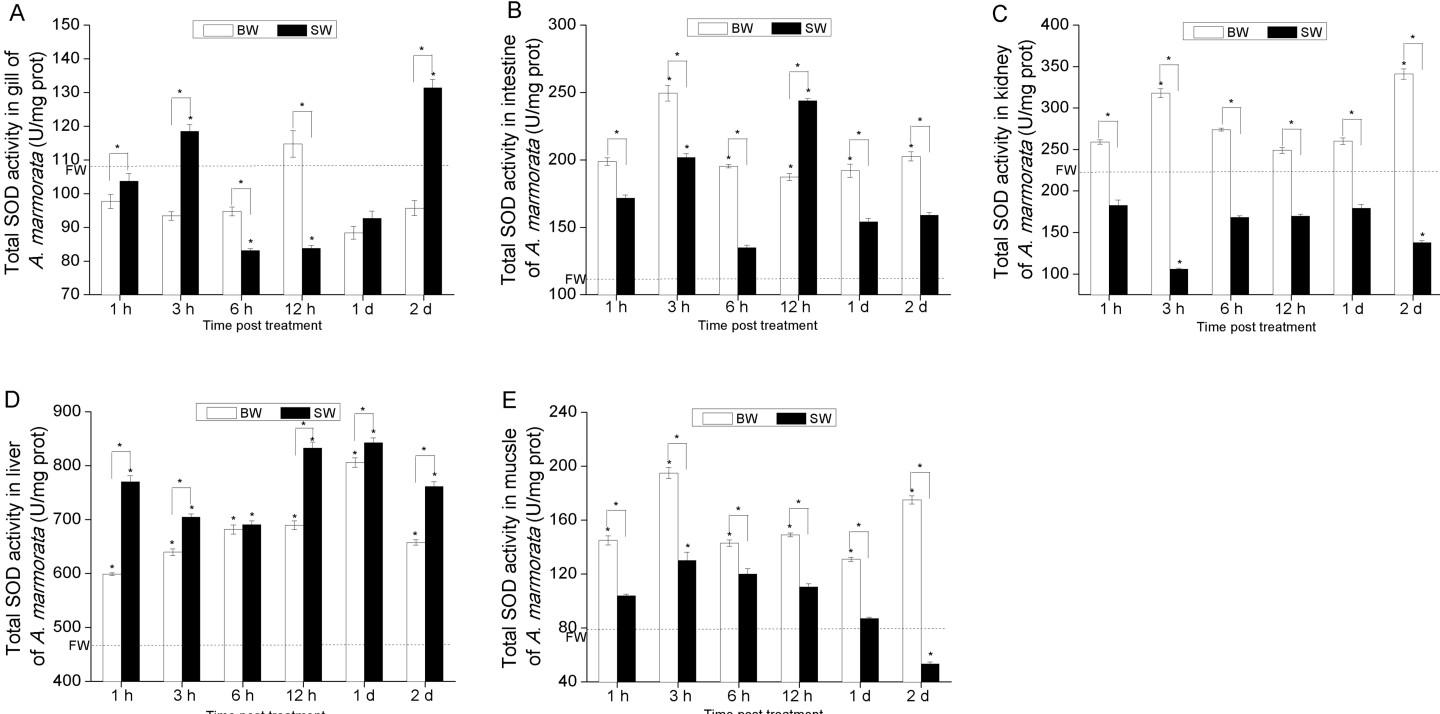

**Figure 4** The total SOD enzymatic activity (*U/mg protein*) analysis of *A. marmorata* in gill (A), intestine (B), kidney (C), liver (D) and muscle tissues (E) of juvenile eels in response to BW and SW adaptation. Vertical bars represent the S.D. ($n = 3$). Data indicated with asterisk symbol (*) are significantly different from corresponding FW group and data with asterisk symbol (*) on the box indicated significant difference between BW and SW, * at $P < 0.05$.

of water environment during migration process, namely Freshwater, Brackish water and Seawater (*Lin et al., 2012*). Therefore, we chose three special salinity of 0‰, 10‰ and 25‰ as the representation of the FW, BW and SW respectively, and we used the oxidative and antioxidant relative tissues, such as liver, kidney, gill, intestine and muscle, to investigate the redox capacity of MnSOD and Cu/ZnSOD of marbled eels in three different salinity range of water environment.

Liver, kidney and intestine have high metabolic rate, which is important to maintain steady-state and normal physiological function for fish, and liver is a vital organ for detoxification and xenobiotic metabolism (*Lushchak, 2015*; *Sun et al., 2014*). Some previous studies suggest that multiple oxidative reactions and antioxidant defenses also occur in gill and muscle tissues (*Ahmad et al., 2006*; *Yin et al., 2011*). A similar tissue distribution profile of AmMnSOD has been reported for the MnSOD in fish *Megalobrama amblycephala* (Yih 1955) (*Sun et al., 2014*) and mollusc *Mytilus galloprovincialis* (Lamarck 1819) (*Wang et al., 2013*), in which a higher expression level is observed in liver and muscle. In addition, the spatial expression pattern of AmCu/ZnSOD was similar to that of Cu/ZnSOD in fish *Pseudosciaena crocea* (Richardson 1846) (*Liu et al., 2015*) and fish *Hypophthalmichthys molitrix* (Valenciennes 1844) (*Zhang et al., 2011*). Therefore, the differential expressions of AmSODs in gill, intestine, kidney, liver and muscle tissues

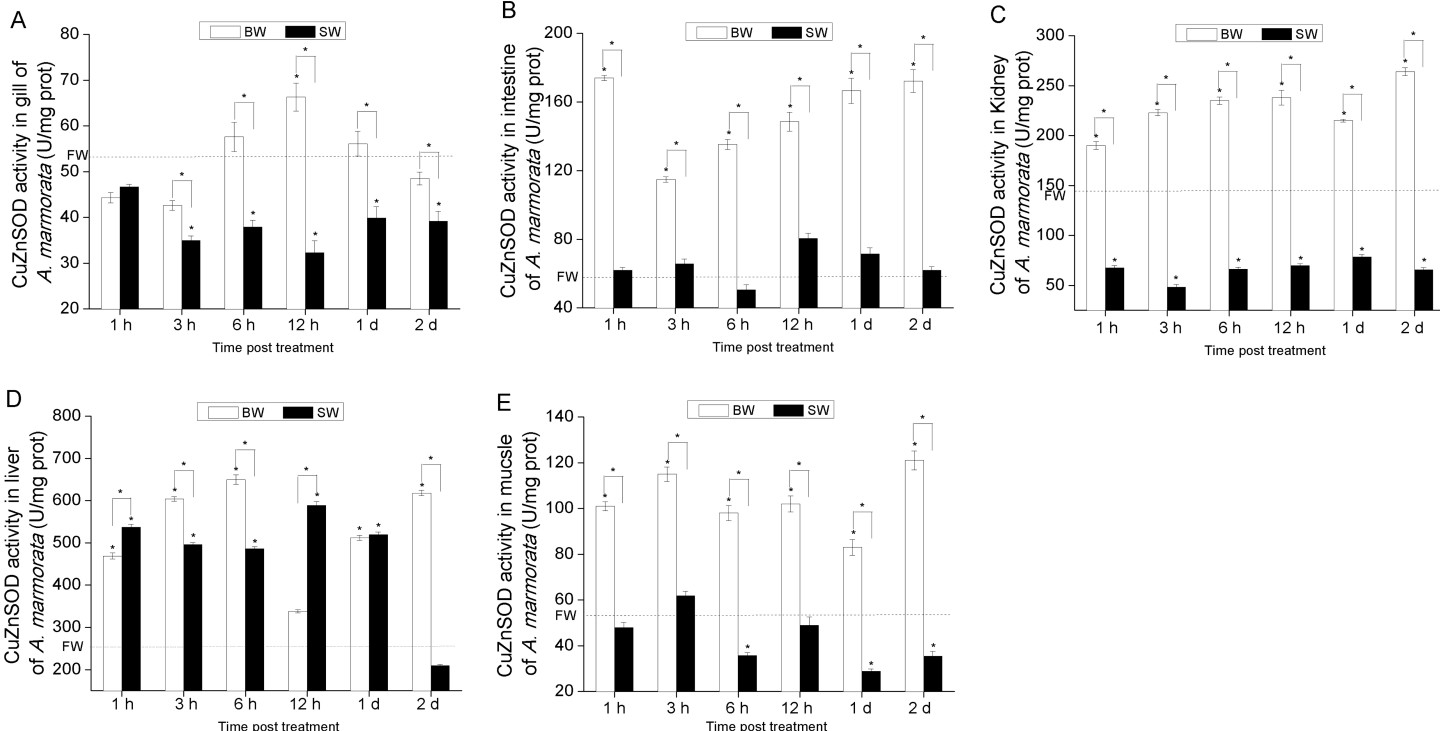

**Figure 5** **The CuZnSOD enzymatic activity (*U/mg protein*) analysis of *A. marmorata* in gill (A), intestine (B), kidney (C), liver (D) and muscle tissues (E) of juvenile eels in response to BW and SW adaptation.** Vertical bars represent the S.D. ($n = 3$). Data indicated with asterisk symbol (*) are significantly different from corresponding FW group and data with asterisk symbol (*) on the box indicated significant difference between BW and SW, ⋆ at $P < 0.05$.

clearly suggested that the expression level of AmSODs was closely related to the antioxidant responses of marbled eels in comparative perspective.

To the best of our knowledge, this is the first report systemically investigating the mRNA expression changes of the two SODs in different tissues following the environment transfer from FW to BW and then to SW. Our results clearly revealed that the salinity variation significantly altered the expressions of AmSODs. Figures 2 and 3 show that both AmSODs were significantly up-regulated in gill, kidney, liver and muscle tissues at early-phases in BW, indicating that these tissues were more sensitive to salt-induced alterations in low level than intestine. Previous study has also shown that liver, kidney and muscle tissues are considered highly susceptible to salinity changes (*Zhao et al., 2008*). However, the expression levels of AmSODs in gill, liver and muscle tissues were inhibited at early-phase in SW, and the AmSOD expression reached its peak at late-phase after acclimatization of 12 h to 1 d. These results suggested that the antioxidant response susceptibility of marbled eels was different when exposed to different salinity stresses. Furthermore, low salinity might stimulate the mRNA expression levels of MnSOD and Cu/ZnSOD, while the high salinity might inhibit their expressions.

Consistent with the changes of AmSODs at the mRNA level in BW, their expressions at the enzymatic activity level appeared to be significantly elevated with the prolonged induction time at the early phase. However, Cu/ZnSOD in liver peaked at 6 h in BW at

the enzymatic activity level, while its mRNA level peaked at 1 h after salinity exposure, suggesting that the relation between enzymatic activity and mRNA expression was not strictly linear, and enzymatic activity is controlled by gene expression as well as enzyme activation (*Chambers & Matrisian, 1997*). The observed lag between the two molecules was probably due to the time difference for *de novo* synthesis of SOD proteins. Previous study has also shown that SOD enzymatic activity is lagged compared with its expression at the mRNA level (*Wang et al., 2016*). Interestingly, the Cu/ZnSOD activity in gill, kidney and muscle tissues was significantly up-regulated in BW, while it was down-regulated in SW compared with FW. These results could be explained by that low salinity stimulates SODs to defend against excessive ROS-induced damage, but their activities may be inhibited once the salinity is above their tolerance range (*Yin et al., 2011*), which partly explains the fatality occurring in juvenile fish *Pampus argenteus* (*Yin et al., 2010*). In addition, in intestine of treated eels, the expression levels of two AmSODs peaked at 12 h in BW and SW, while the total SOD activity was significantly up-regulated at early-phase in BW and SW, indicating the strong antioxidant responses in intestine of eels when exposed to different salinities.

In conclusion, based on the expression profiles of AmSODs at the mRNA and enzymatic activity levels after salinity exposure, we supposed that SODs in juvenile marbled eels could be effectively enhanced by low salinity but inhibited when the salinity was higher than the body tolerance. Also, in the total SOD enzymatic activity and Cu/ZnSOD activity levels, only the SOD activities in liver could keep an up-regulated trend within 2 d in SW, while those in gill, kidney, intestine and muscle tissues were inhibited in varying degrees. Therefore, we inferred that liver might possess stronger redox capacity compared with other tissues.

In the present study, we identified two SODs, denoted as AmMnSOD and AmCu/ZnSOD. Moreover, we assessed their mRNA expression levels in eels in FW and analyzed the temporal mRNA expression profiles and enzymatic activity *in vivo* after they were transferred to BW and SW. All these results indicated that AmMnSOD and AmCu/ZnSOD played vital roles in the adaptation of marbled eels to salinity variation. Moreover, our findings provided new and valuable evidence to further clarify the physiological adaptation and regulatory mechanism of SODs in eels.

## ACKNOWLEDGEMENTS

The authors thank Xiaolu Wang and Shaowu Yin for field and laboratory support.

### Funding

This study was supported by the National Natural Science of Jiangsu Province (BK20141450), the National Natural Science Foundation of China (30770283), and Project Foundation of the Academic Program Development of Jiangsu Higher Education Institution (PAPD). The funders had no role in study design, data collection and analysis, decision to publish, or preparation of the manuscript.

## Grant Disclosures

The following grant information was disclosed by the authors:

National Natural Science of Jiangsu Province: BK20141450.

National Natural Science Foundation of China: 30770283.

Foundation of the Academic Program Development of Jiangsu Higher Education Institution (PAPD).

## Competing Interests

The authors declare there are no competing interests.

## Author Contributions

- Li Wang conceived and designed the experiments, performed the experiments, analyzed the data, wrote the paper, prepared figures and/or tables, reviewed drafts of the paper.
- Xiaolu Wang conceived and designed the experiments, performed the experiments, analyzed the data, wrote the paper, reviewed drafts of the paper.
- Shaowu Yin conceived and designed the experiments, contributed reagents/materials/-analysis tools, reviewed drafts of the paper, provided funding and supervision.

## Animal Ethics

The following information was supplied relating to ethical approvals (i.e., approving body and any reference numbers):

All eels were collected from Wenchang, Hainan Province, China by Hainan Wenchang Jinshan Eel Technology Co., Ltd. This company obtained the People's Republic of China aquatic wild animal catching permit from Ministry of Agriculture of The People's Republic of China in 2004 (Approval number: National Fishery Resources and Environmental Protection 2004; 13). This study was also approved by the Ethics Committee of Experimental Animals at Nanjing Normal University (Research permit number: NNU20120301).

## DNA Deposition

The following information was supplied regarding the deposition of DNA sequences:

GenBank accession numbers: KR350467–KR350468.

## Data Availability

The raw data was supplied as Supplementary Files.

## Supplemental Information

Supplemental information for this article can be found online at http://dx.doi.org/10.7717/peerj.2149#supplemental-information.

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
