# Peer review of "Effects of salinity change on two superoxide dismutases (SODs) in juvenile marbled eel Anguilla marmorata"

_PeerJ, doi:10.7717/peerj.2149_

## Round 0.1 · original submission · Major Revisions

Please see the reports from the 3 referees. You need to improve the clarity of your manuscript by, among other things, changing the presentation and description of the results.

Reviewer 1 ·

Basic reporting

The context of some background is unclear.
Lines 59-62: the literature regarding the connection between salinity and antioxidant responses of fish should be briefly reviewed to show what have been known and what the knowledge gaps were left.

Experimental design

1. the rationals for the research question is not convincing.
Lines 71-72: the proposed question “it remains unclear how SODs are regulated….” does not scientifically sound. The authors are suggested to propose a scientific question based on some rationales or mechanism background in terms of the importance or the physiological significance in salinity adaptation of the species.

2. Figures 2 and 3 are the most important data in the present study; however, the experimental designs and the data presentation neither scientifically nor technically sound. SODs in the species appear to be very sensitive to environmental salinities; and therefor appropriate control and enough repeats are important and necessary to precisely clarify the effects. All the data of BW and SW were normalized by that in FW at 0 h, and this is difficult to follow and of no physiological significance. If there are no data in FW at different time points, FW group is not an appropriate control. Comparisons should be made only between BW and SW at each time points, and between different time points at the same salinity. Furthermore, the data with only 3 repeats (n = 3) for each treatment are not convincing at all.

Validity of the findings

1. The discussion should be accordingly re-written after the data set are appropriately revised.
2. Figs 2 and 3 have 2 factors, salinity and time. Two-way ANOVA with pairwise comparisons should be conducted.
3. The present study is related to the adaptation to environmental salinity. There are no any data to show the physiological performances after salinity challenges. Another weakness is that the whole discussion does not take the physiological mechanisms in different organs for salinity adaptation into consideration.

·

Basic reporting

I made the comments altogether. Please follow the same

Experimental design

It would have been better,

Validity of the findings

OK.

Additional comments

The article “Effects of salinity change on two superoxide dismutases (SODs) in juvenile marbled eel Anguilla marmorata” by Wang et al. is a nice piece of work which is dedicated to deteremine the redox response of the eel in he form of SODs levels a altered salinities. The English language meets the international standard for the article. I strongly recommend he article to be published in Peer J. However, I urge that for common readership and for improvement of their article, the following points be considered by the authors.
Abstract
There are many redox regulatory molecules. So, I suggest the authors may dedicate a line why did they choose SOD only.
“We found that different tissues of juvenile eels possessed varied salinity tolerances, and liver exhibited the largest tolerance range. “ Tolerance to what? In relation to death/survival? How the tolerance range of the fish to salinity is justified by measuring the SOD level. Tissue content of SOD is a response or redox capacity to superoxide radical level (which may be modulated by salinity or other environmental conditions). May be all the related lines be modified as the redox response level of the fish in the form of SOD content with respect to salinity level.
Key words: “Redox response” may be added
Introduction:
Nicely structured but following points needs to be modified.
SODs do not dismutate all ROS. So its substrate name must be written instead of ROS as common.
All SODs have not similar rather same role i.e. to dismutate superoxide radicals. So, the first line of the of second paragraph needs to be changed.
“Theoretically, Cu/ZnSOD and MnSOD are localized in cytoplasm and mitochondrial matrix, respectively.” Is not true always. It is just reverse in some of the aquatic organisms. Please replace Theoretically with Usually.
….a bulk scavenger of superoxide radicals…
I do not think change in salinity can be a pollutant as mentioned in second line of the third paragraph.
….paid to draw the connection between salinity and antioxidant responses in fishes.
The first and second lines of the fourth paragraph may be modified as Marbled eel Anguilla marmorata belongs to Osteichthyes, is one of the quintessential tropical catadromous fishes. This fish live widely across tropical and subtropical oceans and are associated with fresh water (FW) systems.
No need to abbreviate IUCN. Is famous, all know it.
…..eels in various aquatic environments.
AmMnSOD and AmCu/ZnSOD may be explained when they were first explained. Did you identify it for the first time, if so ok, otherwise please modify “ we identified”
MATERIALS AND METHODS
This section sounds good.
“EXPERIMENTAL EELS“ is redundant.
After acclimatized in our laboratory for 1 week …….., experimental eels (but in which condition?? What salinity, and other condition for normal and experimental condition?) In introduction, its ambient salinity condition may be explained.
If the treatment period is 7 days then, it would have been increased after doing pilot experiments for their survival and external lively activity.
The treated eels were primarily placed in FW (0 h, salinity of 0‰)….If it is 0 h then why you mention you placed in FW.
The treated salinity for 1, 3, 6, 12, 24 and 48 h could have been changed to days instead of hours.
………cDNA SYNTHESIS
“salinity of 0.85%” is redundant.
I think the authors should explain clearly about the exposure of animals to what condition like they explained clearly about the protocols. I am still confused how many groups are there in the whole experiment.
For SOD activity measurement the authors should always use proper buffer system with the required osmotic pressure and anti proteases.
RESULTS
Statistical significance level must be indicated in figure 1. If they used SPSS v17.0 software, then p value can be mentioned in text.
DISCUSSION
Discussion is clear, short, impressive and understandable. The authors compare their results with many organisms. I suggest for general readership the common name such as fish/mollusc/crab etc. can be put before the scientific names. Because it will be difficult to understand from the scientific name at glance for a reader with whom the comparison has been made. There are few other articles published between 2010 to 2016 are also there which dedicated to know the tissue specific levels of SODs at altered salinities in aquatic animals. They may be included in discussion.
SODs play a crucial role in eliminating excessive superoxide radicals and….. damage in organisms.

Reviewer 3 ·

Basic reporting

No Comments

Experimental design

No Comments

Validity of the findings

No Comments

Additional comments

General Comments to Authors
Interesting study, but needs revision to improve clarity. Two of the 3 figures are too crowded and the descriptions of them are lengthy. Look to simplify presentation. Discussion also needs work to focus on significance of results.

Specific Comments to Authors
Introduction
Line 50 the authors write, “SOD expression is regulated by endotoxins …” I don’t think “regulated” is the right word here.

Line 59 & 60 – It would be useful to give a brief recap about what is know about the connection between salinity and antioxidant responses.

Is there any particular reason the authors chose the two SODs to study?

Line 74 – the phrase “under normal conditions,” what is “normal?” Do the authors mean “in FW?”

Methods
Were eels collected from FW?

Line 134& 135 – SOD activity was measured “after the salinity change” is vague. Do the authors mean after the target salinity was reached?

Results
Figures 2 & 3 are extremely busy and hard to read. There are too many small graphs. A better way to organize and present the results is needed. Do the authors need to present all tissues? At the very least perhaps the BW and SW graphs could be presented separately.

The descriptions of Figs 2 & 3 are dense and must be simplified and clarified. Each graph is slightly different and is described in detail. Perhaps overall trends could be described or only typical graphs (if any) only are presented.

Discussion
Line 213 – I’m not sure what “undulation of salinity” and “psychological stress responses” means. Do the authors mean “physiological stress responses?” In general manuscript should be reviewed for proper use of English language.

The discussion should focus on the significance of the authors’ findings. The main points seem lost. The authors must make sense of all the graphs of expression and activity and present a central message

Overall, it would have been useful to take blood samples at the time of sampling to measure plasma ion levels or osmotic pressure. It would have given some indication about the severity of stress.

---

## Round 0.2 · Minor Revisions

Your manuscript still needs more clarity as per the comments of Reviewer 3.

·

Basic reporting

OK.

Experimental design

OK

Validity of the findings

OK

Additional comments

Thank you for revising the MS as per the suggestions of the reviewers. I encourage you to continue your work on this particular subject. Still some minor English typo and grammatical mistakes exist (such as in abstract :other issues" will be "other tissues") and please take care of those.

Reviewer 3 ·

Basic reporting

See General Comments

Experimental design

See General Comments

Validity of the findings

See General Comments

Additional comments

General Comments to Authors
Improved, but still needs work. Description of results is better, but still could be clearer. Discussion needs to focus on significance of results. Both would benefit from revision that focuses on clarity. Data seem “noisy” to me and authors must decide what the data are telling them given this.

Specific Comments to Authors
Introduction
Line 40 – “As an indicator of the water environment” can be omitted at the beginning of the sentence.

The first time ROS is mentioned (line 42) it should be spelled out.

Line 53 – Beginning of sentence “have not similar rather same role” makes no sense.

The authors cite a study on a fish, Pampus argenteus, that is exposed to low salinity. I presume it is a marine fish. This information should be given.

Methods
The sampling regime, which is described in lines 106 & 107, should be given earlier (lines 101).

Why was total SOD activity measured? How does total SOD activity relate to the specific SODs you were focused on.

Results
Results is improved, but still too focused on describing each graph. At the very end of the “Temporal Transcriptional Regulation of the two AMSODS” section the authors take a step back and write, “viewing all these tissues as a whole.” I think they should present all their results. What were the major trends that were observed? This perspective gets lost when focusing on how transcript rose in hour 1, but fell in hour 3 only to rise again in hour 12.

I wonder about the jump in transcription rates during the first hour when followed by several hours of low rates. Are these spikes real?

Description of the methods indicate it took about 3 days to go from FW to BW and about 8 days to go from FW to SW. I wonder if the lower transcription rates in SW reflect a time effect, not a salinity effect. Perhaps the response has largely passed by the time the fish get to SW. Isn’t it likely that the stress response began as soon as salinity was raised at the start and continued for the whole time that salinity was being raised. Making measurements only after the target salinity was reached may miss much of the response.

Figs. 4 & 5 - It is difficult to distinguish between the BW and SW lines.

I don’t really know what to make of the enzyme activity data (Figs 4 & 5). Some tissues/salinities are well below FW levels while others are well above. Further, I see no consistent trends that indicate an induced response that follows the up-regulation of transcription. Rather activities often fluctuate widely from one sampling period to the next. Given these issues it is difficult to draw conclusions about the connection between transcription and enzyme activity.

Discussion
It is always better to organize the discussion around the significance of your results, not the literature. I suggest revision of the discussion with this in mind. Further given the limitations of the data, the authors should consider what their data unambiguously tell them about SOD transcription and activity with salinity challenge.

Several times in the discussion the authors refer to “the variation trend.” I don’t know what this means.

---

## Round 0.3 · Minor Revisions

We expect the authors to add discussion on the possible impact of time difference for salinity transfers (3 days for BW and 8 days for SW) on their results.

Reviewer 3 ·

Basic reporting

See below

Experimental design

See below

Validity of the findings

See below

Additional comments

General Comments to Authors
My main concerns with the previous draft was the presentation of results and the discussion of the data. I see only marginal improvement here.

Specific Comments to Authors
Results
What is a “crosscurrent trend” (line 187)?

Discussion
Discussion should start on line 221. That is when the authors begin to discuss their results.

The authors did not really respond to my criticism about their methodology. They took 3 days to raise the salinity from FW to BW, and they took 8 days to raise the salinity from FW to SW. Yet they treat the SW fish as if they did not respond to any salinity changes until the SW salinity as reached. In fact, the fish were probably responding all along. It seems to me that this must be factored into their evaluation of the fish’s response. Yet it is never mentioned.

---

## Round 0.4 · accepted · Accept

After considering your revisions, your manuscript is now acceptable for publication.